# Production, Physicochemical and Structural Characterization of a Bioemulsifier Produced in a Culture Medium Composed of Sugarcane Bagasse Hemicellulosic Hydrolysate and Soybean Oil in the Context of Biorefineries

**Fernanda Gonçalves Barbosa [1], Paulo Ricardo Franco Marcelino [1], Talita Martins Lacerda [1]**, **Rafael Rodrigues Philippini [2], Emma Teresa Giancaterino [2], Marcos Campos Mancebo [1], Júlio Cesar dos Santos [1] and Silvio Silvério da Silva [1,\***

[1] Department of Biotechnology, Engineering School of Lorena, University of São Paulo (EEL/USP), Lorena 12602-810, SP, Brazil
[2] Bosque Foods GmbH, 10435 Berlin, Germany
\* Correspondence: silviosilverio@usp.br; Tel.: +55-1231595308

**Abstract:** Biosurfactants are amphipathic molecules, biodegradable, with reduced toxicity. They can be synthesized by fermentative processes from oleaginous compounds and agro-industrial by-products. In this context, the present study describes the production and the physical, chemical, and structural characterization of the bioemulsifier secreted by the yeast *Scheffersomyces shehatae* 16-BR6-2AI in a medium containing hemicellulosic sugarcane bagasse hydrolysate combined with soybean oil. The bioemulsifier was produced in Erlenmeyer flasks and isolated; then, the physicochemical and structural characterization of the formed molecule was carried out. The following fermentation parameters were obtained: $Y_{X/S}$ = 0.45, $Y_{P/S}$ = 0.083, and productivity of 0.076 g/L/h. The bioemulsifier was found to be a polymer containing 53% of carbohydrates, 40.92% of proteins, and 6.08% of lipids, respectively. The FTIR spectrum confirmed the presence of functional groups such as amides, amines, and carbonyls. The bioemulsifier was stable over a range of temperature ($-20\,°C$ to $120\,°C$), salinity (1–15%), and pH (2–12). It was observed that the biomolecule has a better emulsifying action in organic solvents with a non-polar character. Therefore, this biomolecule is a potential substitute for synthetic surfactants and can be used in different applications.

**Keywords:** bioemulsifier; sustainability; lignocelulosic biomass; hemicellulosic hydrolysate; soybean oil; biorefineries

## 1. Introduction

In recent decades, there has been an intense global transition from a petroleum economy to a bioeconomy [1]. As a result, the demand is increasing for known sustainable or eco-friendly products. In the context of biorefineries, these products are produced from biomass. They have properties such as high biodegradability, low toxicity, and more biocompatibility. Therefore, they can be used by society to cause a less negative impact on the environment [2,3].

Among the important products of modern society and those which are obtained by petroleum derivatives, surfactants stand out for their various applications. These compounds are used in many industrial sectors, from fine chemicals to pharmaceuticals, agricultural products, and oil recovery [4,5]. However, the use of synthetic surfactants causes serious environmental problems. These include foaming in rivers and increased concentrations of xenobiotic compounds such as PCBs (polychlorinated biphenyls) and PAHs (polynuclear aromatic hydrocarbons) that can be derived from synthetic surfactant structures. The presence of these compounds causes a reduction in the oxygen level of the

water, deterioration of aquatic ecosystems, and disturbance of the water cycle [6]. Thus, environmentally friendly biosurfactants synthesized by microorganisms in biorefineries have been highlighted in research and industries to be used in cosmetic, food, pharmaceutical, and agricultural applications.

Biosurfactants are natural metabolites obtained from a great diversity of bacteria, filamentous fungi, and yeasts by fermentation processes [7,8]. In addition to presenting better performance in extreme conditions of temperature, pH, and salinity, these molecules also have reduced toxicity, high biodegradability, and greater biocompatibility [9,10]. Among the main microorganisms that produce biosurfactants, there are bacteria such as *Bacillus* and *Pseudomonas* and yeasts such as *Candida*, *Starmerella*, and *Scheffersomyces*. Studies indicate that yeasts have an advantage over bacteria due to their GRAS (generally recognized as safe) status, in addition to having cellular structures more resistant to the biosurfactants secreted and accumulated in cultures/culture medium compared to certain types of bacteria [11,12].

Like synthetic surfactants, biosurfactants have amphipathic structures and emulsifying and surface-active properties. These properties are related to the structural characteristics and classifications of these compounds. They are classified into molecules of high molecular weight and low molecular weight. Low molecular weight biosurfactants, such as glycolipids, lipopeptides, and phospholipids, are more effective in reducing surface tension. High molecular weight compounds are called bioemulsifiers (BE) and include lipoproteins, lipopolysaccharides, and polymers (emulsan, alasan). These act in the stabilization of emulsions. [13,14].

The structure of biosurfactants/bioemulsifiers depends on the microorganisms that produce them, and on the substrates used as a carbon source during fermentation processes [8,15]. The production of these compounds has the advantage of using renewable substrates as nutrients. Among them, there are substrates used in different biorefineries, such as oilseeds, saccharines, and starches [12,16,17]. Lignocellulosic biorefineries, composed of lignocellulosic biomass, such as sugarcane bagasse and straw, are commonly used for the production of ethanol from cellulose. Due to their chemical composition, these biorefineries have the potential for the production of biosurfactants. Microorganisms capable of metabolizing C-5 carbohydrates can be used in bioprocesses that use the hemicellulosic fraction of lignocellulosic biomass as a substrate [8,18]. Furthermore, oleaginous substrates, such as soybean oil, canola, and hydrocarbons, are used in the synthesis of biosurfactants/bioemulsifiers. Examples include the production of lipopeptide biosurfactant by *Streptomyces* sp. in a medium supplemented with palm oil, and residual car oil as a substrate for the chemically mutated *Bacillus subtilis* strain [19,20]. Studies suggest that the combination of two carbon sources with different polarities favors the production of BE by reducing foam formation during the production process, in addition to producing biomolecules with better physicochemical properties [9,14,15].

Thus, this study aimed to produce and characterize the BE produced by the yeast *Scheffersomyces shehatae* 16-BR6-2AI in a medium containing sugarcane bagasse hemicellulose hydrolysate combined with soybean oil in the context of biorefineries. This work will contribute to the development of sustainable alternatives obtained by the integration of the sugarcane and biodiesel biorefineries, aiming to use raw materials involved in these sectors to obtain value-added products.

## 2. Materials and Methods

### 2.1. Materials

The chemical reagents used in this study included potassium dihydrogen phosphate, ammonium nitrate, magnesium sulfate heptahydrate, hydrochloric acid, sulfuric acid, sodium hydroxide, D-(+)-xylose (≥99%), D-(+)-glucose (≥99.5%), yeast extract, malt extract, bacteriological peptone, granular activated charcoal, hexane, toluene, cyclohexane, ethyl acetate, phenol, petroleum ether, chloroform, deuterated water, and Folin–Ciocalteu reagent. They were purchased from SigmaAldrich (St. Louis, MO, USA). All chemical reagents were

of analytical reagent grade. Commercial kerosene, lubrication oil, soybean oil, sunflower oil, and corn oil, were purchased from local market brands.

### 2.2. Preparation and Characterization of Sugarcane Bagasse Hemicellulosic Hydrolysate

Sugarcane bagasse was kindly donated by Usina Ipiranga (Descalvado, SP, Brazil). The acid hydrolysis was performed in a 30 L stainless steel reactor, loaded with sugarcane bagasse and sulfuric acid in a solid/liquid ratio of 1:10 at 121 °C for 15 min [21]. After hydrolysis, the sugarcane bagasse hemicellulosic hydrolysate (SBHH) (liquid fraction) was concentrated in a 30 L vacuum evaporator at 70 °C until xylose concentration reached approximately 90 g/L. Subsequently, a fraction of the concentrated hydrolysate was detoxified using an overliming technique combined with the addition of 1% activated charcoal [22]. Both hydrolysate (concentrate non-detoxified and concentrate detoxified) were quantified in relation to the content of monosaccharides, acetic acid, and furans by high-performance liquid chromatography (HPLC) [23], and phenolics compounds by spectrophotometric methods [24]. They were kept at −20 °C to prevent possible degradation and contamination processes.

### 2.3. Maintenance of Microorganisms

The strain *Scheffersomyces shehatae* 16-BR6-2AI was obtained from the Collection of Social Insects Study Center, Rio Claro Biosciences Institute, São Paulo State University (UNESP—Universidade Estadual Paulista, Rio Claro, SP, Brazil). The culture was maintained on media yeast malt agar (YMA) (g/L: glucose, 10; yeast extract, 3; malt extract, 3; and bacteriological peptone, 5), and after 48 h of static growth in a bacteriological incubator at 30 ± 2 °C, the cultures were transferred to a refrigerator at 4 °C.

### 2.4. Inoculum Preparation with the S. shehatae 16-BR6-2AI

The inoculum was prepared using a modified Kitamoto medium (g/L: potassium dihydrogen phosphate, 2; yeast extract, 1; ammonium nitrate, 20; magnesium sulfate heptahydrate, 2; and pure xylose, 40). The pH of the media was adjusted to 5–5.5 [25].

For inoculum preparation, one loopful (3 mm) of the yeast strain was transferred to a 125 mL Erlenmeyer flask containing 25 mL of the modified Kitamoto medium. The flask was incubated at 200 rpm at 30 °C for 48 h in a rotary shaker. After incubation, the cells were isolated by centrifugation at $12,000 \times g$ for 10 min, and resuspended in 0.9% (*w/v*) sterile saline solution. Approximately 1 mL of cell suspension was inoculated into the fermentation flasks to obtain an initial absorbance of 1.0 at 600 nm, corresponding to $1 \times 10^7$ total cells/mL.

### 2.5. Bioemulsifier Production by S. shehatae 16-BR6-2AI

The BE production was performed in triplicate in 1 L Erlenmeyer flasks containing 200 mL of a medium composed of (g/L) 6.0 peptone, 5.0 ammonium nitrate, 40.0 xylose in SBHH, and 100.0 (*v/v*) soybean oil. The flasks were incubated at 30 °C and stirred at 200 rpm for 144 h using a rotary shaker. After fermentation, the cultures were centrifuged at $12,000 \times g$ for 10 min to remove the cells. The cell-free medium was washed with P.A hexane to remove the oil. The emulsifying index ($EI_{24}$) and surface tension (ST) of the BE-containing supernatant were measured (Section 2.12). The supernatant was then used for the isolation and characterization procedures. At the end of the fermentation, the parameters of productivity (YP/S and YX/S) of the produced BE were calculated.

### 2.6. Bioemulsifier Isolation Produced by S. shehatae 16-BR6-2AI

Extraction of BE was evaluated using four different solvents: ethyl acetate [26,27]; chloroform:methanol (4:1) [28]; chloroform [29], and ethanol [30].

In the extraction with ethyl acetate, the cell-free medium of the fermentations was acidified with HCl to pH 2.0 and kept under refrigeration at 4 °C for approximately 12 h. After this period, liquid–liquid extraction was carried out. This involved adding ethyl

acetate to the acidified supernatant (in three series), shaking in a separatory funnel, and discarding the aqueous phase. For quantification, the organic phase was evaporated, and the weight of the mass formed was determined [27].

In the extraction with chloroform:methanol (4:1) and with chloroform, the cell-free medium was placed in a separatory funnel and then the respective solvent was added. After the addition of the solvent, it was stirred until the complete separation of the phases was achieved, and then the aqueous phase was discarded. Lastly, the organic phase was evaporated and the mass of the BE was determined [28,29].

For ethanol extraction, ice-cold ethanol (92–96%) was initially added to the cell-free medium and kept under refrigeration at 4 °C for approximately 18 h. After this period, the sample was centrifuged at $12,000 \times g$ for 20 min at 4 °C, and the formation of a precipitate was observed and washed with chloroform:methanol (2:1). Then, the suspension was centrifuged, and the precipitate formed was resuspended in ethanol. This phase was evaporated and the weight of the extracted BE was determined [30]. The selection of the extraction method was based on yields (where a higher concentration of BE was extracted) and $EI_{24}$ values (Section 2.12).

After selecting the extraction method, the crude BE was dialyzed in water for 24 h, using a 12 KDa membrane, then the sample was lyophilized. The crude and lyophilized BE was employed in the steps of physicochemical characterization and tests were carried out to evaluate the $EI_{24}$ as described in Section 2.12.

### 2.7. Biochemical Characterization of the Bioemulsifier Produced by S. shehatae 16-BR6-2AI

Total carbohydrates, proteins, and lipids were quantified by spectrophotometry (Spectrophotometer Jenway 7305, Staffordshire, United Kingdom). Total sugars were dosed according to the Dubois et al. [31] methodology using a mixture of phenol–sulfuric acid as the chromogenic reagent and glucose as the standard for the preparation of the calibration curve at 490 nm. Total proteins were quantified according to the Lowry et al. [32] methodology using Folin reactive as a reagent and the bovine serum albumin (BSA) solution as the standard protein solution. The number of proteins can be estimated via reading the absorbance at 750 nm. Total lipids were quantified using phosphoric acid vanillin reagent and read in absorbance at 530 nm [33].

All tests were performed in triplicate.

### 2.8. Physicochemical Characterization of the Purified and Lyophilized Bioemulsifier Produced by S. shehatae 16-BR6-2AI

#### 2.8.1. Molecular Weight

The molecular weight of the BE was determined by gel permeation chromatography (GPC, Shimadzu, Kyoto, Japan) with two Phenomenex Polysep-SEC GFC-P columns [34]. The column was calibrated with standard polyethylene oxide (PEO) and the elution was monitored using a Shimadzu refractive index detector (Kyoto, Japan). The elution was performed using a mobile phase formed by 50 mM phosphate buffer solution pH 7.0 containing 0.2 M sodium azide and 0.05 M potassium nitrate filtered through a PVDF filter membrane (0.45 um).

#### 2.8.2. Elemental Analysis of Carbon (C), Hydrogen (H), Oxygen (O), and Nitrogen (N)

Quantitative elemental analysis of the BE (10 mg) to determine the percentage of different elements (CHON) present in the sample was carried out using PerkinElmer PE 2400 equipment (Waltham, MA, USA).

#### 2.8.3. Fourier Transform Infrared Spectroscopy (FTIR)

FTIR analysis of the BE was carried out using a spectrometer (DX Perkin Elmer, Waltham, MA, USA) with potassium bromide (10 mg of sample and 90 mg of potassium bromide) as support in a wavenumber range of 4000–400 cm$^{-1}$ [17,35].

### 2.8.4. Nuclear Magnetic Resonance ([1]H NMR and [13]C NMR)

1H NMR and 13C NMR analysis was performed with the purified BE in Bruker equipment (500 MHz) (Bruker, Germany) operating at 25 °C using $D_2O$ as solvent [36].

### 2.8.5. Gas Chromatography and Mass Spectroscopy

The BE was analyzed on a gas chromatography–mass spectrometry system (CG–MS QP2010 Ultra SHIMADZU, Kyoto, Japan) equipped with CarbolWAX and a 5% methyl–phenyl silicone column. The initial column temperature was 50 °C for 1 min then increased to 280 °C (10 °C/min). To identify the lipid content, acid hydrolysis of the purified BE was performed. Approximately 10 mg of the sample was mixed with 1 mL of 5% HCl:methanol solution overnight at room temperature. After the reaction was quenched with water (1 mL), the methyl ester derivatives of the fatty acids were removed with n-hexane [37].

### 2.8.6. X-ray Diffraction

X-ray diffraction measurements were carried out with an Empyrean diffractometer (PANalytical, Almelo, The Netherlands) equipped with an X'Celerator X'Pert detector. A Cu anode was used as the X-ray source (K radiation: 40 kV and 40 mA), and a 1/4° divergence slit was used to collect the data in the 2θ range of 10–80° [17].

### 2.8.7. Thermal Analysis of the Bioemulsifier

The determination of the thermal stability of the BE for thermos gravimetric analysis (TGA) was performed using NETZSCH STA 449F3 Jupiter (Selb, Germany). Fourteen milligrams of the sample was placed onto the platinum pan and heated from 30 °C to 700 °C at a rate of 10 °C/min under a nitrogen atmosphere. The thermostability of the BE was measured as the percentage of weight loss in the stated temperature range (30–700 °C). Phase transitions were performed by differential scanning calorimetric (DSC) analysis. An amount of 5.2 mg of the same sample was placed in the sample pan of the DSC Q20 brochure–TA Instruments and heated constantly at 10 °C/min to 550 °C [36].

### 2.8.8. Zeta Potential

Measurements of zeta potentials were performed with isolation BE at room temperature in a ZetaSizer Nano ZS (Malvern Panalytical, Kassel, Germany). The BE samples were diluted in water and subjected to analysis, which was performed in three reading cycles in automatic mode (with duration undetermined) [28].

### 2.9. Effect of Bioemulsifier Concentration on the Stability of the Emulsion

The effect of concentrations of BE on the stability of the emulsion was determined by mixing different concentrations of BE with kerosene. BE solutions with different concentrations (100; 90; 80; 70; 60; 50; 40; 30; 20; 10; 5; 2.5; 0.5; and 0.1 mg/mL) were prepared in Falcons tubes. The tubes were mixed by vortexing at 9000 rpm for 2 min and incubated at room temperature for 24 h. The emulsion was maintained for $EI_{24}$ analysis (Section 2.12). The tests were performed in triplicate.

### 2.10. Turbidity Measurement

The turbidity test was performed according to Ohadi et al. [38]. The turbidity of the prepared BE solutions of different concentrations (100; 90; 80; 70; 60; 50; 40; 30; 20; and 10 mg/mL) was measured by using a UV/Vis spectrophotometer (Shimadzu, UV-2550, Kyoto, Japan) at room temperature. The reported values corresponded to the absorbance at a wavelength equal to 600 nm.

### 2.11. Evaluation of Emulsifying Properties in Different Hydrophobic Substrates, Temperatures, pH Values, and Salinity

The emulsifying properties of the BE produced by *S. shehatae* medium were evaluated in different hydrophobic substrates, such as oils (mineral oil, corn oil, sunflower oil, and

soybean oil) and organic solvents (hexane, cyclohexane, phenol, ethyl acetate, chloroform, toluene, kerosene, and petroleum ether). The hydrophobic substrates were mixed with 2.0 mL of a solution containing BE extracted at 40 mg/mL. The tubes were mixed by vortexing at 9000 rpm for 2 min and were incubated at room temperature for 24 h.

The effect of thermal stability, pH, and salinity on the emulsifying properties of BE were evaluated by mixing 40 mg/mL of the extracted BE solution and kerosene. For thermal stability, emulsions were prepared and kept at determined temperatures (ranging from −20 to 121 °C) for 15 min, and then kept at room temperature. To evaluate the influence of pH, its values were adjusted using 1.0 mol/L NaOH or HCl in extracted BE solution before mixing it with kerosene. The effect of salinity was considered with different concentrations of monovalent salt (NaCl 5–15% *w/v*) until complete dissolution [39]. The concentration of NaCl in the BE solution was adjusted, and then the emulsions were formed. The emulsion was maintained for $EI_{24}$ analysis (Section 2.12). The tests were performed in triplicate.

### 2.12. Emulsifying and Tensioactive Properties of Bioemulsifier Produced by S. shehatae 16-BR6-2AI

The emulsifying index ($EI_{24}$) was determined by the addition of 1 mL of kerosene to the same volume of BE solution in glass tubes. The tubes were mixed by vortexing at 9000 rpm for 2 min and were incubated at room temperature for 24 h. The stability of the emulsion was determined after 24 h, and the $EI_{24}$ was calculated as described previously by [40]. Surface tension (ST) was determined using a SensaDyne QC6000 tensiometer (Mesa, AZ, USA).

### 3. Results and Discussion

#### 3.1. Characterization of Sugarcane Bagasse Hemicellulosic Hydrolysate

The xylose presented in the SBHH was used in this study as one of the carbon sources for BE production by yeast *S. shehatae* 16-BR6-2AI. At the end of the hydrolysis process with dilute acid, 18 g/L of xylose was obtained. In this study, the hydrolysate was concentrated by vacuum concentration. Due to the drastic conditions of pH and temperature of the hydrolysis process, in addition to the solubilization of fermentable sugars, the formation and release of compounds inhibiting microbial growth (phenolics, furans, and acetic acid) were observed [41]. To reduce these inhibitors, the SBHH underwent a detoxification process using the precipitation method by pH variation and adsorption with activated carbon. Thus, the concentrated and detoxified SBHH used in the fermentative process had $94.89 \pm 0.05$ g/L of xylose, $18.75 \pm 0.05$ g/L of arabinose, $9.12 \pm 0.07$ g/L of glucose, besides $4.05 \pm 0.2$ mg/L of total phenolics and $118.0 \pm 7.0$ mg/L of furans (furfural and HMF). It was observed that acetic acid, furans, and phenolic compounds were found in smaller amounts compared to sugars. In the literature, few works describe the influence of furans and phenolic compounds on the production of BE by yeasts. The study carried out by Yu et al. [42] observed a lower production of sophorolipids by *Starmerella bombicola* in a medium supplemented with corn husk hydrolysate without detoxifying. The sophorolipid yield obtained with the medium composed of detoxified hydrolysate was 2.07 times higher than that obtained with the non-detoxified medium. Marcelino et al. [17] observed the production of glycolipid by *Scheffersomyces stipitis* from detoxified hemicellulose hydrolysate of sugarcane bagasse. This hydrolysate contained about $0.028 \pm 0.006$ and $0.91 \pm 0.003$ mg/L of furans (furfural and HMF) and total phenolics, respectively.

#### 3.2. Bioemulsifier Production by S. shehatae 16-BR6-2AI in SBHH Combined with Soybean Oil

The yeast *Scheffersomyces shehatae* is a microorganism of the genus *Scheffersomyces* sp. This microorganism is generally considered versatile, unconventional, and a model organism mainly for the production of bioethanol. In addition, this microorganism can metabolize C-5 (xylose) and C-6 (glucose) carbohydrates [23,43,44]. In 2019, Marcelino et al. [23] described the production of biosurfactant by the yeast *S. shehatae* 16-BR6-2AI in a medium containing SBHH. The biosurfactant produced showed preferential emulsifying activity, with an $EI_{24}$ of 50%, in addition to larvicidal activity against *Aedes aegypti* [23].

In the current study, the production of BE by the yeast *S. shehatae* 16-BR6-2AI was carried out in a medium combining two important substrates commonly used in lignocellulosic and biodiesel biorefineries. The culture medium used combined xylose present in SBHH (40 g/L) and soybean oil (100 g/L).

After the fermentation process was determined from the cell-free medium, the surface tension reduction and the emulsifying capacity were evaluated by $EI_{24}$. From the analyses, it was observed that the BE present in the cell-free medium was able to emulsify kerosene with an $EI_{24}$ (%) of 62.5 ± 2.05. However, the surface tension reached values of 58.9 mN/m, and therefore, this biomolecule acted preferentially as an emulsifying agent, being less effective in the reduction of the surface tension. This property is dependent on the size and structure of the molecule, and compounds with high molecular weight have greater effectiveness as an emulsifying agent. This property is evaluated according to its ability to maintain a stable emulsion of at least 50% after 24 h of its formation [35,45]. In the study of Da Silva et al. [46], BE produced by *Yarrowia lipolytica* by using residual glycerol as a carbon source, had a surface tension that averaged 41.7 mN/m and an $EI_{24}$ that reached 56%.

After the experiments verified the production of BE by *S. shehatae*, some quantitative parameters related to the consumption of substrates and microbial growth were determined. In the present work, BE production occurred in a culture medium supplemented with SBHH containing about 40 g/L of xylose and 100 g/L of soybean oil, and an inoculum of 0.1 g/L of cells. After 144 h of fermentation, the total consumption of xylose, 8 g/L of residual soybean oil, the formation of 60 g/L of cells, and the production of 11 g/L of BE were observed. Thus, the following fermentation parameters were obtained: YX/S = 0.45, YP/S = 0.083, and productivity of 0.076 g/L/h. It is noteworthy that the values of the fermentation parameters obtained in the present work do not represent optimized conditions of the bioprocess. Therefore, the optimization of this process can be carried out in the future, aiming at the increment of these parameters for the scaling of the process.

### 3.3. Isolation of Bioemulsifier Produced by S. shehatae 16-BR6-2AI in SBHH Combined with Soybean Oil

For the isolation of BE, different extraction methods were evaluated to verify, among the solvents tested, whether ethanol could be considered the best solvent for the extraction of BE, as reported in the literature. Four solvents were tested, including ethyl acetate, chloroform: methanol (4:1) mixture, chloroform, and ethanol [26–30]. According to the literature, these solvents are the most used in the extraction of BS/BE of glycolipids, glycoproteins, lipoproteins, and polymers [47].

The determination of BE concentration was performed by gravimetry. After the extraction process, the extracted BEs were weighed and their concentrations in g/L were determined. Then, these samples were suspended in distilled water, in the same initial volume as the extracted sample. With the BE solution, the $EI_{24}$ and the reduction of surface tension were evaluated (Table 1).

**Table 1.** Data from the different extraction methods evaluated for BE produced by *S. shehatae* 16-BR6-2AI in medium based on hemicellulosic hydrolysate and soybean oil.

| Solvents | $EI_{24}$ (%) | T.S (mN/m) | BE (g/L) |
|---|---|---|---|
| Chloroform: methanol | 48.3 ± 0.33 [a] | 61.9 | 7.4 ± 0.07 [a] |
| Chloroform | 47.6 ± 0.26 [a] | 60.1 | 6.5 ± 0.06 [b] |
| Ethanol | 58.3 ± 0.33 [b] | 63.9 | 7.8 ± 0.07 [c] |
| Ethyl acetate | 48.3 ± 0.32 [a] | 63.5 | 7.7 ± 0.08 [c] |

Where: $EI_{24}$ (%)—emulsion index 24 h; T.S (mN/m)—surface tension; BE (g/L)—BE concentration. [a–c] Different letters indicate significant differences according to Tukey's test ($p < 0.05$).

It was observed that the BE produced did not promote an effective reduction in surface tension. Therefore, to choose the extraction method, the amount of BE and the $EI_{24}$ values obtained in each extraction were considered. According to the data in Table 1, higher

BE yields were obtained through extraction by ethanol precipitation and by liquid–liquid extraction with ethyl acetate. When performing Tukey's statistical test, no significant difference was observed in the yield obtained between these two methods. With regard to the emulsifying property, determined by the $EI_{24}$, a higher $EI_{24}$ was observed in the emulsions formed by the BE extracted by ethanol (58.33%) (Figure 1a). Therefore, for the isolation of BE produced by *S. shehatae*, the method selected was ethanol precipitation. By this method, the formation of a precipitate was observed, and after the evaporation of the solvent, a white powder was formed (Figure 1b). As shown in the works by Monteiro et al. [30] and Pasa [48], this precipitate is indicative of the presence of BE in the extracted medium.

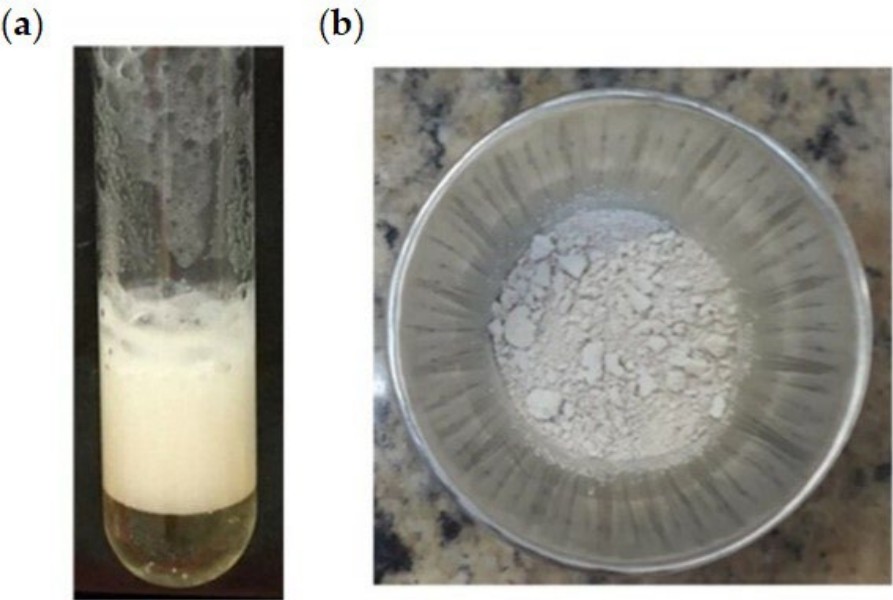

**Figure 1.** Bioemulsifier extracted with ethanol. (**a**) Emulsion formed with the bioemulsifier extracted by ethanol; (**b**) dry mass of bioemulsifier after solvent evaporation.

The difference observed between the concentrations of BE (g/L) and the $EI_{24}$ obtained in the different solvents (Table 1) is probably due to the difference in the polarity of the solvents and their interactions with the BE molecules. Among the solvents used in this study, chloroform has a more nonpolar character (polarity index 4.1), performing interactions of the induced dipole-induced dipole type with the most nonpolar regions of the BE. Ethyl acetate has intermediate polarity (index equal to 4.4) and is less nonpolar than chloroform. Its polar functional group interacts dipole–dipole with the polar regions of BE, since the carbonic chain, with a nonpolar character, interacts with nonpolar regions of the molecule. The solution formed by the chloroform: methanol mixture has an intermediate polarity, having nonpolar and polar characteristics. In this case, there is a strong interaction between the hydrophobic groups of the BE from chloroform and the carbonic chain of methanol, in addition to the interaction between the polar groups of the BE and the hydroxyl group of methanol. Ethanol, on the other hand, has a greater polar character (polarity index equal to 5.2) due to the presence of the OH- (hydroxyl) group, promoting strong interaction between the hydrogen bond of ethanol and the polar groups of the BE. However, the carbonic chain of this compound gives it a nonpolar characteristic, making this solvent capable of interacting with nonpolar regions of the BE.

Ethanol precipitation is a process that promotes the reduction of the dielectric constant of the medium, which promotes the aggregation of molecules by charge interaction [49]. Commonly, in extraction methods using solvents, toxic and expensive solvents are used. Therefore, following the concepts of green chemistry, it is necessary to use cheap and less toxic solvents for the recovery of BEs. The main advantage of the ethanol precipitation

method is that this solvent has zero toxicity, in addition to being recoverable, which facilitates its use in an industrial process [47].

The choice of the method to be used in the extraction followed the physical–chemical and structural characterization of the BE. With the biomolecule extracted, the process of dialysis in water was carried out, using a 12 KDa membrane, to remove residual sugar molecules and salts. Then, with the extracted and dialyzed sample, it was lyophilized to carry out the tests for biochemical and physicochemical characterization.

### 3.4. Biochemical Characterization of Bioemulsifier Produced by S. shehatae 16-BR6-2AI in SBHH Combined with Soybean Oil

The biochemical characterization indicates that the BE produced by *S. shehatae* 16-BR6-2AI in a medium containing SBHH and soybean oil consists of a carbohydrate–protein complex with low lipid content. According to data obtained, the molecule contained 53% of carbohydrates, 40.92% of proteins, and 6.08% of lipids, therefore, it presented a polymeric structural characteristic. Other works such as the one carried out by [50] showed that the BE produced by *Yarrowia lipolytic*, in a medium composed of glucose, consisted of 45% carbohydrate, 47% protein, and 8% lipid. Bhaumik et al. [51] produced, from *Meyerozyma caribbica*, a proteoglycan composed of 56% carbohydrate, 20% protein, and 2% lipid. Another BE composed of proteoglycan was obtained by *Acinetobacter junii* with 50.5% protein, 43% carbohydrate, and a small fraction of lipids, about 3.8% [52]. From the biochemical characterization of these molecules and of the BE presented in this study, these compounds consisted of a polymeric complex formed by carbohydrate–protein–lipid with emulsifying properties. The combination of these biochemical groups in BE molecules increases its emulsifying capacity [45,53].

### 3.5. Physicochemical Characterization of Bioemulsifier Produced by S. shehatae 16-BR6-2AI

The molecular mass of the BE produced by the yeast *S. shehatae* was evaluated by GPC. The data indicated that this molecule has an average molecular mass (Mw), estimated at 26.376 Da, and a polydispersity of 1.8. This value is similar to other compounds characterized as BE, such as liposan, which has a molecular mass of 27.600 Da [54], in addition to the BE proteoglycan produced by *Meyerozyma caribbica*, with a value of 34.000 Da [51].

As shown by the biochemical analysis, the BE produced is formed by a polymeric complex, and its average molecular weight was similar to high molecular weight BS. In addition, elemental analysis was performed, and it was determined that the molecule has the following proximate formula: C57.8% H9.1% O30.7% N2.4%.

Based on these results, an analysis of infrared spectroscopy (FTIR) was performed to identify its main functional groups (Figure 2). The absorption spectrum in the infrared region showed a band between 3200.0–3600.0 $cm^{-1}$, a characteristic region of hydroxyl (-OH) and amine (-NH) stretching; however, the amine stretching band was probably camouflaged by the -OH band, and bands commonly found in the carbohydrates and peptides portion [35]. At 2928.0 and 2853.0 $cm^{-1}$ there are elongation bands, characterizing symmetric and asymmetric (C-H) stretching of the -CH2 and -CH3 of aliphatic chains [16]. Furthermore, the absorption peak at 1741.5 $cm^{-1}$ characterizes a C=O carbonyl bond, a characteristic peak of aliphatic chain esters, common in lipid esters [38,55]. The 1605.0 $cm^{-1}$ band represents a C=C vibration band associated with proteins as well as a C=O stretching band of amides. The presence of an absorption peak at 1414.0 $cm^{-1}$ is related to the stretching of N-H into amide [28,55]. These peaks are usually found in polysaccharides and proteins. Furthermore, the band at 1013.5 $cm^{-1}$ characterizes the presence of stretching C-CO-C group bonds, commonly found in glycolipids [35]. Similar FTIR spectra for molecules with emulsifying characteristics have been reported for bioemulsifiers synthesized by *Lactococcus lactis* [56], *Lactobacillus pentosus* [57], *Corynebacterium kutscheri* [58], *Stenotrophomonas maltophilia* [59], and *Burkholderia thailandensis* [35].

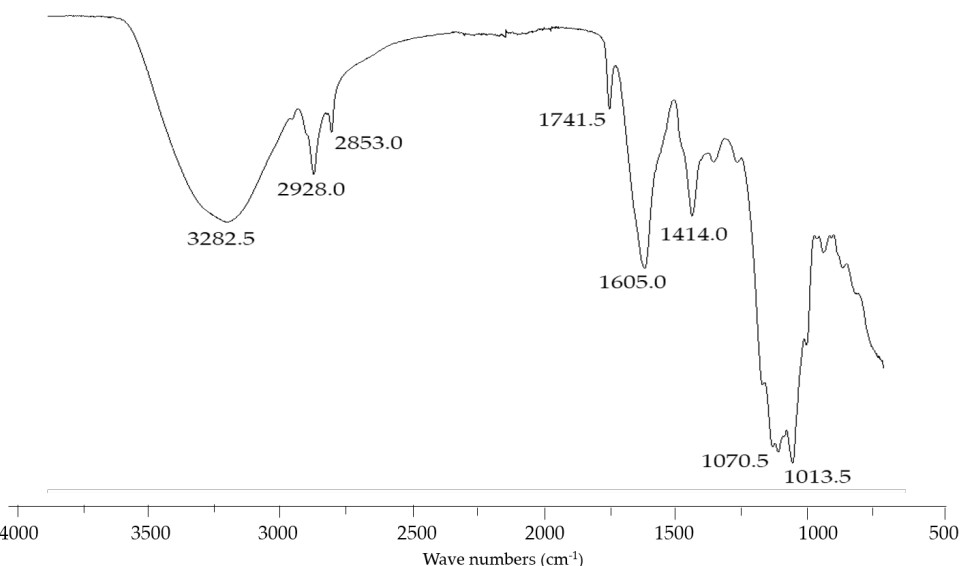

**Figure 2.** Infrared spectra of crude BE scanned in the range 500–4000 cm$^{-1}$.

According to 13C NMR analysis, signals were observed in the region between 16.78–71.01 ppm typical of saturated carbons (sp3) not linked to electronegative elements, probably from methyl (CH$_3$), methylene (-CH$_2$), and methine (-CH-) groups. These groups are common in hydrocarbon chains present in lipids, proteins, and carbohydrates [60]. The region between 71.01–80.70 ppm are signals of functional groups that present electronegative atoms, such as O and N [61]. They are probably C-O and C-N bonds of carbonyls, carboxyls, ethers, esters, and amides present in carbohydrates, proteins, and lipids. The peaks above 100 ppm are probably from unsaturated carbons with double bonds (C=C) and carbonyls of amides, acids, esters, aldehydes, and ketones. Although it was not possible to elucidate the complete structure of the molecule due to its complexity, the 13C NMR data confirms the presence of functional groups in carbohydrates, lipids, and proteins, which corroborates the data obtained from the FTIR and spectrophotometric analyses.

In the H NMR analysis, signals were observed at 2.1–2.5 ppm, which may be related to peaks adjacent to the carboxyl carbonyl group (-CH-COOH); to hydrogen peaks at the α position next to the N of an amide group (-CH-CONH-); or to hydrogen peaks adjacent to carbonyls in aldehydes (R-CH-CH=O) and ketones (R-CH-C(R)=O). The region between 0.5–1.8 ppm may refer to the hydrogen of the hydroxyl group. The peaks for this group are highly variable, their position depending on concentration, solvent, and temperature; the peak can be widened at its base by these factors. In the 0.9–1.3 ppm signals, the peaks can also refer to the methyl (CH$_3$), methylene (-CH$_2$), and methine (-CH-) groups [38,62]. In long carbon chains, the -CH$_2$- absorptions can overlap in a single unresolved peak. These peaks refer to hydrogens linked to functional groups common to lipids, carbohydrates, and proteins.

The H NMR analysis for complex molecules can reveal data that allows for only the partial characterization of the compound since overlapping peaks are a common occurrence due to the multiple effects of shielding and deshielding by anisotropy. As seen in the 13C NMR analysis, the results obtained in the present analysis also confirmed the presence of functional groups typical of proteins, carbohydrates, and lipids.

Analysis of the profile of fatty acids in the lipid chain present in the BE was performed by GC–MS. According to the data obtained in this analysis, the present biomolecule has in its constitution palmitic acid, stearic acid, oleic acid, linoleic acid, and 13-docosenoic acid.

This BE molecule is also formed by a protein region, and an X-ray diffraction analysis was performed to verify the presence of crystalline regions. In Figure 3, peaks are observed indicating the presence of crystalline and amorphous regions. The region with an organized crystalline structure was shown by intense peaks in the range of 14.74–26.39°.

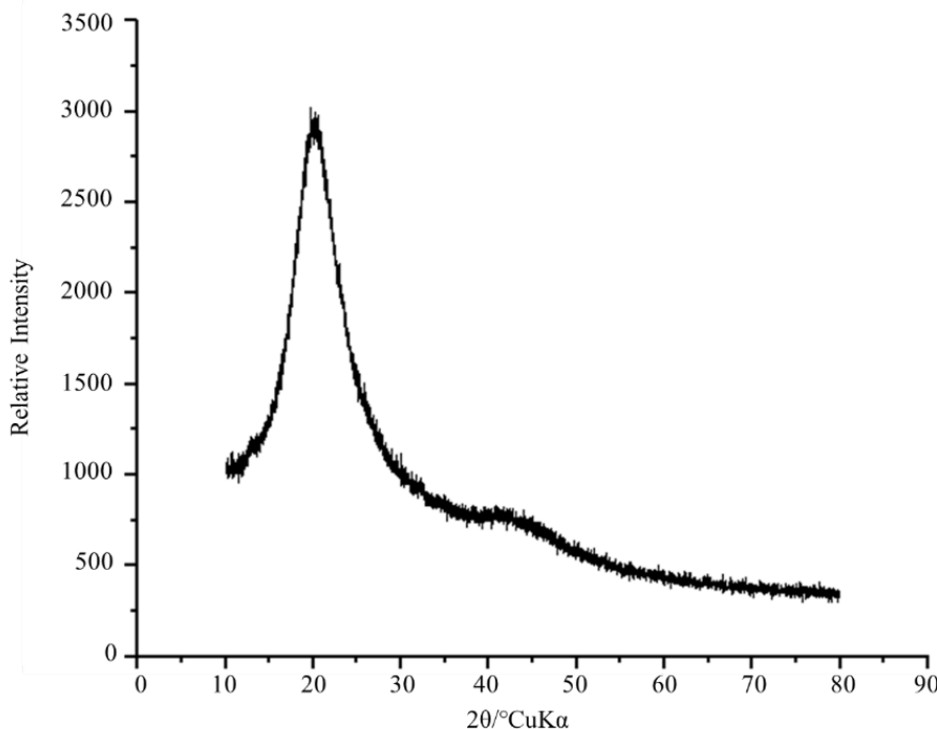

**Figure 3.** XRD profile of the crude BE produced by strain *S. shehatae* 16-BR6-2AI. Examined with 2θ in the range 10–90°. The diffractogram represents the sharp diffracted peaks of the crystalline portion and the broad peaks of the amorphous part.

Thermostability analysis of BEs allows us to verify the possible applications that these molecules can play. Thermogravimetric analysis was performed in the range of 30–700 °C. As seen in Figure 4, for the BE produced there were three points of thermal degradation. Initially, a mass loss of approximately 2.34% was observed at 150–170 °C. This loss can be attributed to the dehydration of the sample. With increasing temperature, the mass remained constant until the temperature reached 348 °C. From that point on, mass loss began to occur, which is likely associated with the decomposition of proteins and polysaccharide side chains. In this temperature range, between 348–450 °C, a mass loss of 55% was detected. At temperatures above 450.0 °C, a gradual loss of mass was observed, which can be attributed to the degradation of residual material. Similar behavior was observed for the BE produced by *Pseudomonas putida* NRRL B-14875, composed of 19.5% (*w/w*) lipids, 21.9% of proteins, and 11.5% of carbohydrates. Initial dehydration occurred between 35.0–103.5 °C, with a weight loss of 5.1%. At temperatures between 283.6–460 °C a mass loss of 46.76% was observed. After this, temperatures between 470–675 °C, caused a mass loss of 15.5% [63].

The BE DSC thermogram shown in Figure 5 revealed a thermal phenomenon close to 68 °C and an accentuated thermal transition within the temperature range of 200–210 °C. At initial temperatures, the occurrence of thermal events is common, possibly due to the evaporation of solvents or some compounds present in the sample. As previously described, the BE from this study was previously isolated from the fermentation medium by ethanol precipitation, so it is suggested that at 68 °C the evaporation of residual ethanol molecules could be present in the sample. The melting temperature (Tm) was observed at 201.61 °C, which suggests the decomposition of the components present in the BE.

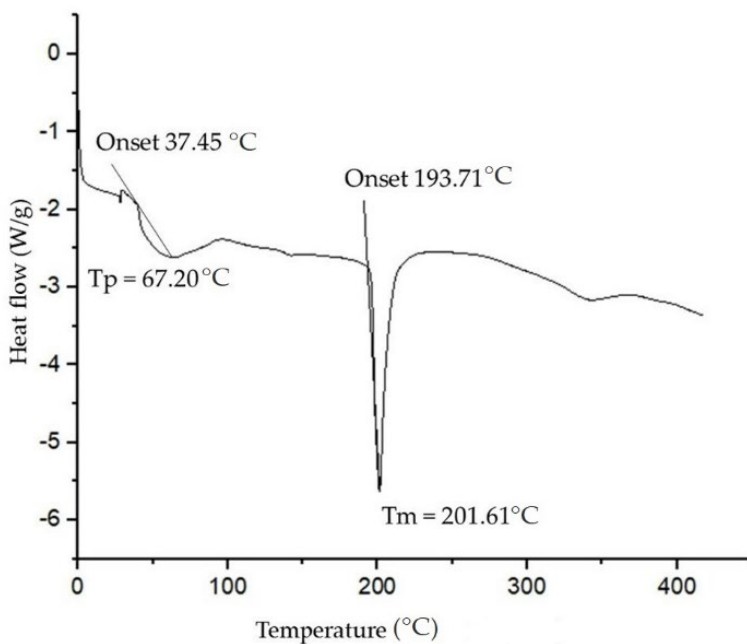

**Figure 4.** Thermogram of the crude BE produced by strain *S. shehatae* 16-BR6-2AI showing thermal stability.

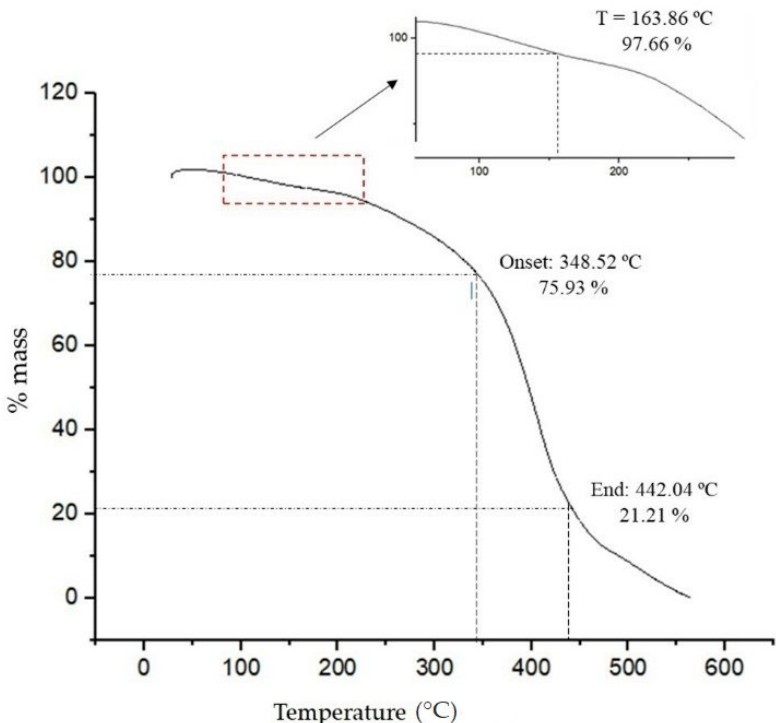

**Figure 5.** DSC profile of crude BE produced by strain *S. shehatae* 16-BR6-2AI showing the thermal transition.

In the study carried out by Ramani et al. [64] for lipoprotein produced by *Pseudomonas gessardii*, DSC analysis showed the thermal transition between 120–150 °C. This transition was attributed to the folding of the protein portion of the biosurfactant, leading to the formation of aggregates with compact mass below this temperature. In the work by Saranya et al. [65], the lipoprotein obtained by *B. subtilis* showed an accentuated thermal transition at 95.44 °C, representing the volatilization of the components of the biomolecule.

The calorimetric analysis information about the thermal stability of the samples was gathered, which is an important property to determine their characteristics when used in extreme temperatures [34,36]. The results obtained suggest that the BE produced from *S. shehatae* is thermostable and can be used in different applications that require heating to obtain a product.

After calorimetric analysis, the study of the zeta potential of aqueous solutions containing BE at 40 mg/mL was carried out. The zeta potential describes the electrostatic potential of the surface of the particles and can indicate the total charge of the molecule [66]. The solution containing the BE at pH 7.0 showed an approximate zeta potential of −29.9 mV. Therefore, it is suggested that the BE produced has anionic characteristics. Measuring this property also provides information on the degree of electrostatic repulsion between particles in a solution. Helping to understand the reasons for the phenomena of coalescence, aggregation, or flocculation in an emulsion, it can be applied to obtain formulations of stable colloidal dispersions [63,67].

### 3.6. Studies of the Emulsifying Properties of the Bioemulsifier Produced by S. shehatae

3.6.1. Evaluation of BE Concentration in the Emulsifying Property

In the emulsification process, surface-active compounds promote stabilization between the dispersed phase and the continuous phase of the emulsion. The BE adsorbs at the oil interfaces, reducing the interfacial tension between the water–oil phases. Thus, it is believed that the emulsifying activity is dependent on the concentration of BE in the emulsifying [68,69]. Therefore, the effect of the isolated BE concentration on the emulsions was evaluated.

Solutions containing different concentrations of BE (100–5 mg/mL) were used in the preparation of emulsions with the hydrophobic phase composed of kerosene. As seen in Figure 6, it was observed that from 20 mg/mL, the emulsions presented an emulsion index above 50%.

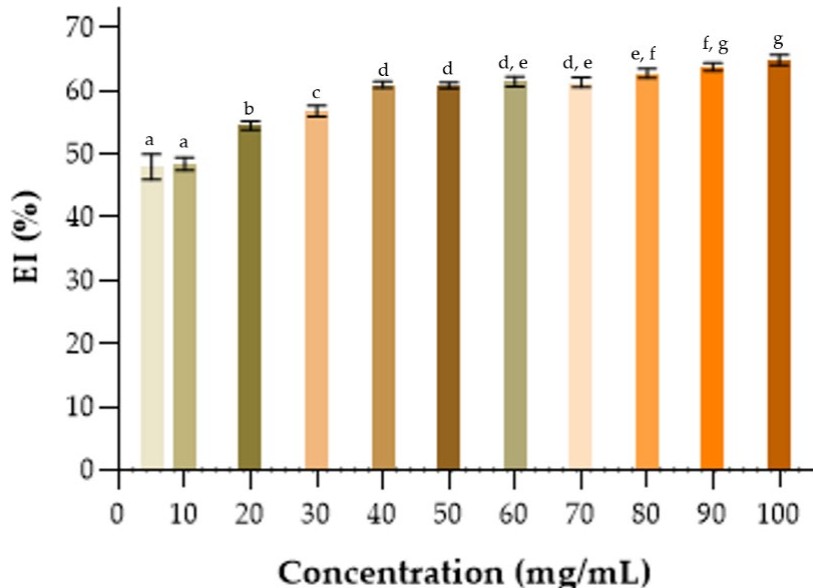

**Figure 6.** Emulsification indices, % $EI_{24}$, of crude BE produced by strain *S. shehatae* 16-BR6-2AI, evaluating the effect of concentrations of crude BE under the emulsion from kerosene. [a–g] Different letters indicate significant differences according to Tukey's test ($p < 0.05$).

It was observed that there was an increase in the emulsifying layer and the stability of the emulsions with the increase in the concentration of BE used. The same effect was observed in the work by Lukondeh et al. [70] and Domingues et al. [71], where there was an increase in the $EI_{24}$ and stability as the BE concentration increased. This saturation

concentration in the emulsifying activity may vary according to the chemical structure of the molecule and other factors, such as the adsorption capacity of the BE and the dispersed phase (e.g., oil) [69,72].

Furthermore, the solubility of BE in an aqueous solution was investigated by the aggregation behavior from the turbidity analysis. BE solutions at different concentrations (10–100 mg/mL) were prepared, homogenized by vortexing for 2 min, and read in a spectrophotometer at 600 nm. Turbidity measurements were performed to study the BE self-association process. Figure 7 shows that BE aggregation behaviors were dependent on the concentration of the BE in the solution. Therefore, an increase in turbidity is related to the increase in BE concentration in the solution [73].

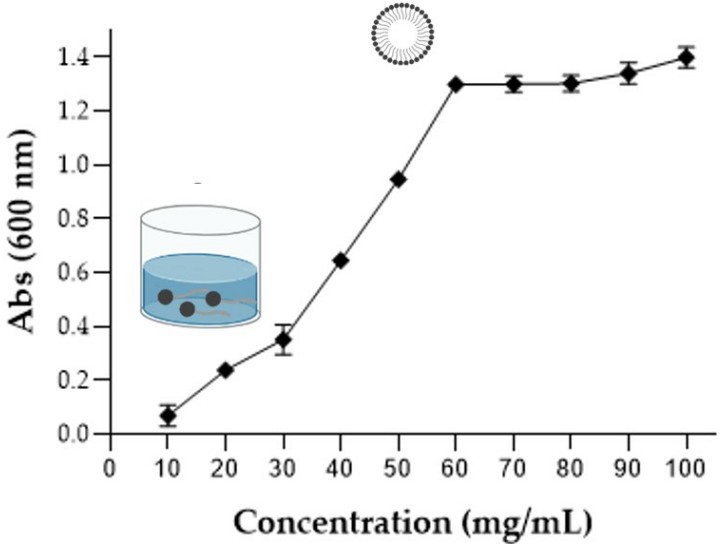

**Figure 7.** Evaluation of aggregate formation by the BE molecules produced by the yeast *S. shehatae* 16-BR6-2AI, by turbidimetry analysis, relating absorbance with BE concentration.

As shown in Figure 7, up to 60 mg/mL there was an increase in turbidity in proportion to the increase in BE concentration, and after that concentration, absorption was stable. Studies have related the increase in turbidity to the size or amount of BE aggregates in the solution [74,75]. The self-aggregation of these biomolecules is related to their ability to form micelles, so it is related to the critical micelle concentration (CMC) of each BE. According to Ohadi et al. [38], the BS/BE may show aggregation behavior above the CMC point [74]. In this study, they found an increase in turbidity for values above the CMC (300 mg/L) of the BE studied. An increase in turbidity was observed up to 500 mg/L, after which there was stability. Based on the statement by Ohadi et al. [38], it is suggested that the CMC of BE produced by the yeast *S. shehatae* varies between 40–80 mg/mL.

Evaluating the concentration of BE as a function of its emulsifying activities and formation of micelle/aggregates, in the analyses carried out later, the use of BE at 40.0 mg/mL was adopted. Above this concentration, there was no increase in the emulsification index and variation in turbidity, which is an indication that this concentration could be close to the CMC of the BE under study. Then, studies were carried out to evaluate the emulsifying capacity of different hydrophobic compounds and their stability in varying temperatures, pH, and salt concentrations.

3.6.2. Evaluation of Emulsifying Properties with Varied Hydrophobic Substrates

The emulsifying capacity of many surface-active compounds depends on their preference for hydrophobic substrates. In this study, the emulsifying property on different hydrophobic substrates using BE alone was investigated. In each glass tube, 1 mL of a hydrophobic substrate and 1 mL of a solution containing the BE at 40 g/L was added. In Figure 8 it is observed that the BE formed emulsions with $EI_{24}$ above 50% for all organic sol-

vents tested, except for petroleum ether and soybean oil. Vegetable oils (corn oil, sunflower oil) and mineral oil formed emulsions with an $EI_{24}$ lower than 50%.

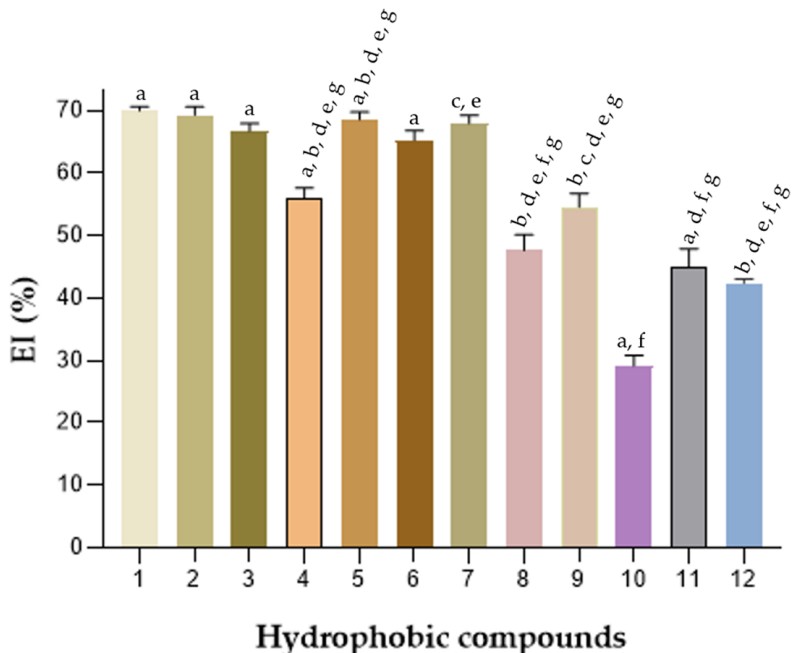

1: Hexane 2: Cyclohexane 3: Phenol 4: Ethyl acetate 5:chloroform 6: Toluene

7: Kerosene 8: Petroleum ether 9: Soybean oil 10: Mineral oil 11: Corn oil

12: Sunflower oil

**Figure 8.** Emulsification indices, % $EI_{24}$, of crude BE produced by strain *S. shehatae* 16-BR6-2AI, against various hydrocarbons and oils. [a–g] Different letters indicate significant differences according to Tukey's test ($p < 0.05$).

The results obtained suggest greater emulsification and stability in emulsions formed by BE with organic solvents, preferably hydrocarbons with more hydrophobic characteristics. As observed in the high $EI_{24}$ obtained for hexane (a hydrocarbon formed only by carbon and hydrogen), chloroform and kerosene, these are highly non-polar compounds. Results similar to those of the present work were obtained by Monteiro et al. [30], Gudiña et al. [61], and Marcelino et al. [23]. All the BE produced proved to be efficient in the emulsification of aliphatic (hexane, chloroform, and ethyl acetate) and aromatic (toluene) hydrocarbons.

According to Rufino et al. [76] and Adetunji et al. [77], the emulsifying capacity directly depends on the affinity between the BE and the hydrophobic substrate. This affinity is related to the composition, chemical structure, and interactions that occur between these molecules. Some factors, such as temperature, pH, and ionic strength, influence the stability of emulsions since they can promote changes in the structure of BE, the property of the media, and the electrostatic forces between the particles. Therefore, stability tests were performed at different temperatures, pH, and ionic strength. As seen previously, the BE produced by the yeast *S. shehatae* proved to be the best emulsifier of the hydrocarbons. Therefore, stability tests were performed with emulsions formed from kerosene.

In the evaluation of the thermal treatment of the emulsions, it may be observed in Figure 9a that the BE was stable at all temperatures evaluated, showing a high $EI_{24}$, above 50% for all temperatures. Even when the BE was subjected to autoclaving temperature, 121 °C, it was still able to form an emulsion with a high $EI_{24}$ value, showing the stability of this molecule to temperature variations. This is an advantage when using these molecules in systems that require thermal treatments, such as some food products, like the preparation of cookies [78].

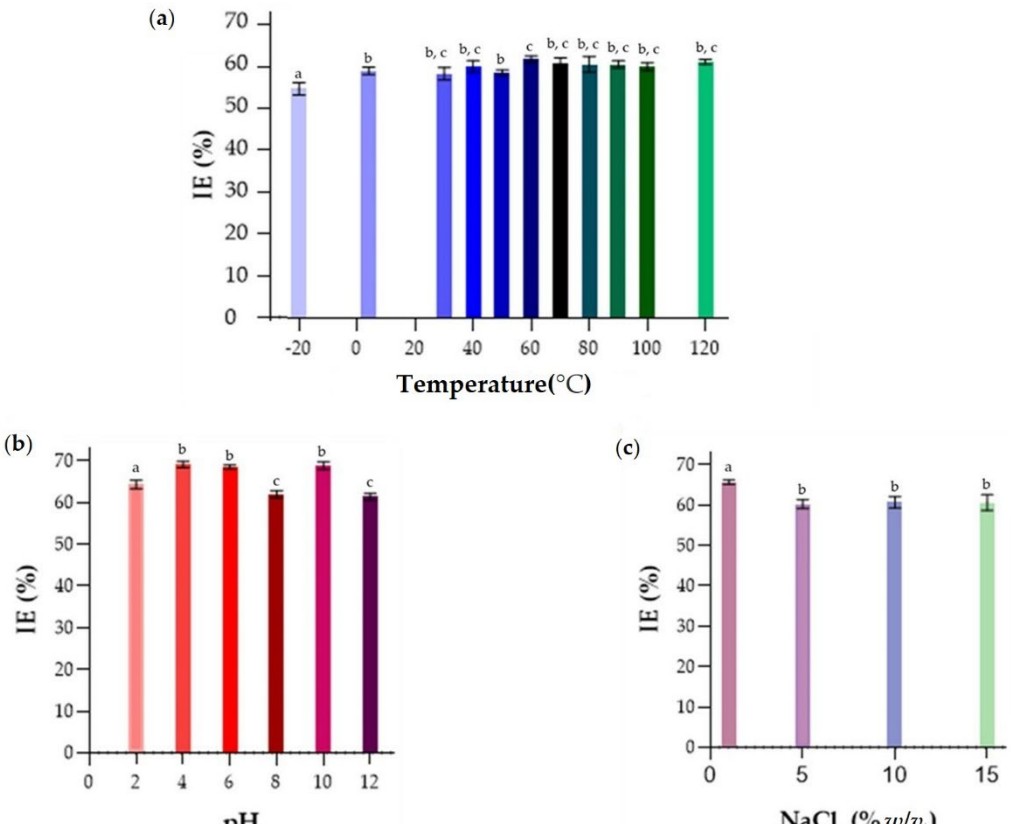

**Figure 9.** Emulsification indices, % EI$_{24}$, of crude BE produced by strain *S. shehatae* 16-BR6-2AI. (**a**) % EI$_{24}$ of BE at different temperatures, (**b**) % EI$_{24}$ of BE at different pH, and (**c**) % EI$_{24}$ of BE at various salinity concentrations. [a–c] Different letters indicate significant differences according to Tukey's test ($p < 0.05$).

The evaluation of the emulsifying capacity between BE and kerosene under different pH (Figure 9b), showed emulsion formation at all pH values studied. After 24 h, the % EI showed values above 60% for all pH. Acid pH values (pH 4), those close to neutrality (pH 6), and pH 10 showed better emulsions. Variations in pH alter the charges present in the medium, alter the solubility of the BE, and affect the structure and size of the micelles formed in the emulsion system [79]. It is likely that extreme pH values caused changes in the molecular structure of the compounds involved in the emulsification process, leading to emulsion destabilization [55]. This result suggests that this biomolecule has the potential to be used in cosmetic products for skin treatments since these products are manufactured and must act at pH 4.5–5.5 [80]. In addition, they can also be used in general cleaning products, and in soaps for washing clothes, where the pH should be 10–12.

In the analysis of the emulsions formed under different concentrations of NaCl (% $m/v$), shown in Figure 9c, the BE studied presented % EI$_{24}$ above 60% and high stability in all applied concentrations. Therefore, it is suggested that this biomolecule can be used in marine environments applied to bioremediation and in other systems where the salt concentration is above the physiological level, as in some processed foods [55]. The increase in saline concentration promotes changes in the electrostatic force of the medium, altering the electrical characteristics of the particles. It can generate attraction or repulsion between the droplets, to promote events such as coalescence or flocculation in the emulsion, subsequently reducing the stability capacity of the emulsions [81]. Thus, it is suggested that when adding 12% ($m/v$) of NaCl to the solution with BE, there has been a change in the electrical charge of this biomolecule, which generated attraction between the particles and a consequent reduction in the emulsifying capacity. Both the addition of an ionic compound

and changes in the pH of the surrounding medium promoted changes in the electrostatic forces of the emulsion, reducing the stability of the emulsions formed.

According to the results, environmental factors such as temperature, salinity, and pH influenced the emulsification process by the BE produced. The BE produced proved to be a sustainable alternative to synthetic surfactants with emulsifying activity. Since the influence of these factors depends on the BE structure, different biomolecules have opposite effects when exposed to different conditions. As in the work by Campos et al. [55], the BE produced by *Candida utilis*, composed of a carbohydrate–protein–lipid complex, showed greater stability under extreme conditions of pH (pH 10), salt concentration (10% $m/v$), and temperature (80 °C). Chandran and Das [82] obtained a biosurfactant produced by the yeast *Trichosporon asahii* and formed emulsions with $EI_{24}$ close to 60% at pH 8. In their study, Luna et al. [83] found a biosurfactant produced by *Candida sphaerica*, with good emulsifying activity in the presence of large amounts of salt. On the other hand, in the study by Rufino et al. [76], the concentration of 12% ($m/v$) of NaCl inhibited the emulsifying action of the biosurfactant produced. In the temperature assessment, the biosurfactant produced by *C. sphaerica* was tested over a wide temperature range and proved to be stable [84]. The same was observed in the work by Marcelino et al. [23], the biosurfactant produced by *Cutaneotrichosporon mucoides* being stable in the applied heat treatment (4–100 °C), and under different concentrations of NaCl (0–10% $m/v$).

The stability of these compounds under different conditions is important when considering the possibility of specific applications. Many industrial products contain a system formed by emulsions in their composition and they must be able to tolerate different formulations, preparations, and storage conditions. In addition to tolerating extreme conditions to be applied in industrial sectors, such as food, pharmaceuticals, cosmetics, and agriculture, among others, these biomolecules must have low or no toxicity.

## 4. Conclusions

The present study reported the production capacity of BE by yeast *Scheffersomyces shehatae* 16-BR6-2AI in a medium containing sugarcane bagasse hemicellulose hydrolysate combined with soybean oil. Characterization tests showed that the molecule produced is a polymeric BE. It excelled in its emulsifying activities in organic solvents and was stable over a range of temperature (−20 °C to 120 °C), salinity (1–15%), and pH (2–12). The results described in these studies are promising because they show the feasibility of the sustainable production of BE by yeasts in integrated biorefineries. The data obtained also showed the potential for the application of this biomolecule in different industrial sectors as an emulsifying agent, from the food industry for products that are treated at varying temperatures to pH-neutral or alkaline cleaning products.

**Author Contributions:** Conceptualization, F.G.B. and P.R.F.M.; methodology, F.G.B., P.R.F.M. and T.M.L.; investigation, F.G.B. and M.C.M.; data curation, F.G.B. and P.R.F.M.; writing—original draft preparation, F.G.B.; writing—review and editing, F.G.B., P.R.F.M., T.M.L., R.R.P. and E.T.G.; visualization, J.C.d.S. and S.S.d.S.; supervision, S.S.d.S. project administration, S.S.d.S.; funding acquisition, S.S.d.S. All authors have read and agreed to the published version of the manuscript.

**Funding:** This research was funded by Brazilian fostering agencies Coordination for the Improvement of Higher Education Personnel (CAPES), Finance Code 001, and São Paulo Research Foundation (FAPESP), grants #16/10636-8 and #16/14852-7.

**Institutional Review Board Statement:** Not applicable.

**Informed Consent Statement:** Not applicable.

**Data Availability Statement:** Not applicable.

**Acknowledgments:** The authors gratefully acknowledge Coordenação de Aperfeiçoamento de Pessoal de Nível Superior–Brasil (CAPES) Finance Code 001, Fundação de Amparo à Pesquisa do Estado de São Paulo (FAPESP), Process number #16/10636-8 and #16/14852-7, and Conselho Nacional de Desenvolvimento Científico e Tecnológico (CNPq) for their financial support. In addition, the authors acknowledge Usina Ipiranga (Descalvado, SP, Brazil) for donating the sugarcane bagasse.

**Conflicts of Interest:** The authors declare no conflict of interest. The funders had no role in the design of the study; in the collection, analyses, or interpretation of data; in the writing of the manuscript; or in the decision to publish the results.

## Abbreviations

| | |
|---|---|
| PCBs | polychlorinated biphenyls |
| PAHs | polynuclear aromatic hydrocarbons |
| GRAS | generally recognized as safe |
| BE | bioemulsifiers |
| C-5 | 5 carbon carbs |
| C-6 | 6 carbon carbs |
| SBHH | sugarcane bagasse hemicellulosic hydrolysate |
| HPLC | high-performance liquid chromatography |
| YMA | yeast malt agar |
| $EI_{24}$ | emulsifying index |
| ST | surface tension |
| PEO | polyethylene oxide |
| HMF | hydroxymethylfufural |
| Mw | average molecular mass |
| Tm | melting temperature |
| CMC | critical micelle concentration |
| FTIR | Fourier transform infrared spectroscopy |
| CG-MS | gas chromatography–mass spectrometry |
| TGA | thermos gravimetric analysis |
| DSC | differential scanning calorimetric |
| $^1$H NMR | $^1$H nuclear magnetic resonance |
| $^{13}$C NMR | $^{13}$C nuclear magnetic resonance |
| YX/S | yield coefficient of cell growth per substrate |
| YP/S | product yield coefficient per substrate |
| BSA | bovine serum albumin |
| GPC | gel permeation chromatography |

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
