# Peer review of "Production, Physicochemical and Structural Characterization of a Bioemulsifier Produced in a Culture Medium Composed of Sugarcane Bagasse Hemicellulosic Hydrolysate and Soybean Oil in the Context of Biorefineries"

_fermentation, doi:10.3390/fermentation8110618_

Round 1

Reviewer 1 Report

The authors intend to investigate the physicochemical characterization of bioemulsifier (SE) of Scheffersomyces shehatae as well as the structure of SE. The authors tried their best to describe the results, most of the time using theoretical aspects. The authors combined the results and discussion together, due to the detailed descriptions of the results and less discussion in depth leading losing focus of the readers. I would rather to see the results and discussion into two separate sections.

The authors also fail to demonstrate the benefits of using bean oil and bagase as the energy sources. Is it more SE production, cost less (how much) especially the bioemusifer activity is not superior than other isolates? 

The authors tried many techniques such as FT-IR, NMR, GC-MS, X ray diffraction, biochemical and many to determine the structure. Unfortunately, the authors did not propose the structure of SE which is expected by the readers.  

There are other techniques issues:

1) Check spacings

2) No abbreviation (BE) in the title or subtitle.

3) All the figure numbers follow the numerological order. Figure 1B --> Figure 1A

4) Move the y axis title to the left . 

Author Response

(1) The authors intend to investigate the physicochemical characterization of bioemulsifier (SE) of Scheffersomyces shehatae as well as the structure of SE. The authors tried their best to describe the results, most of the time using theoretical aspects. The authors combined the results and discussion together, due to the detailed descriptions of the results and less discussion in depth leading losing focus of the readers. I would rather to see the results and discussion into two separate sections.

Thank you for the suggestion. The journal Fermentation allows the authors to choose how to present the results and discussions. Considering the high volume of results and the relationship between them, we decided to integrate all the information for the sake of didactics.

(2) The authors also fail to demonstrate the benefits of using bean oil and bagase as the energy sources. Is it more SE production, cost less (how much) especially the bioemusifer activity is not superior than other isolates?

Thank you for your questions. We conducted a thorough bibliographic research and found many contributions related to the production of biosurfactants/bioemulsifiers in the presence of either water soluble or insoluble carbon sources, but only a few considering both nutritional sources. Studies suggest that the combination of two carbon sources with different polarities favors the production of BE by reducing foam formation during the production process, in addition to producing biomolecules with better physicochemical properties (Amaral et al., 2008; Jimoh; Lin, 2019; Jahan et al., 2020). Herein, we chose not to mention this topic because it is a work in progress in our research group, in which we are evaluating the influence of adding soybean oil and sugarcane bagasse hydrolyzate for the production of BE. The work of Marcelino et al. (2019), however, already indicated that the biosurfactant produced by Scheffersomyces shehatae in a culture medium supplemented only with xylose from the hemicellulose hydrolyzate of sugarcane bagasse had an emulsion index (IE24h) of 48.1%. Herein, we considered the addition of soybean oil to the culture medium and this same yeast produced a biosurfactant with IE24h of 62%. It is possible to infer, therefore, that the presence of soybean oil improved the emulsifying activity and/or modified the chemical structure of the produced molecule. Jahan et al., 2020 stated that changes in the carbon source impact the chemical structure of the corresponding biosurfactant. This information about this relationship were added to Section 1 (Introduction) lines 81 to 83.

Marcelino, P.R.F.; Peres, G.F.D.; Terán-Hilares, R.; Pagnocca, F.C.; Rosa, C.A.; Lacerda, T.M.; dos Santos, J.C.; Da Silva, S. S. Biosurfactants production by yeasts using sugarcane bagasse hemicellulosic hydrolysate as new sustainable alternative for lignocellulosic biorefineries. Ind. Crops Prod. 2019129, 212-223. DOI: 10.1016/j.indcrop.2018.12.001

Jimoh, A.A.; Lin, J. Enhancement of Paenibacillus sp. D9 lipopeptide biosurfactant production through the optimization of medium composition and its application for biodegradation of hydrophobic pollutants. Appl. Biochem. Biotechnol. 2019187, 724-743. DOI: 10.1007/s12010-018-2847-7

Jahan, R.; Bodratti, A.M.; Tsianou, M.; Alexandridis, P. Biosurfactants, natural alternatives to synthetic surfactants: Physicochemical properties and applications. Adv. Colloid Interface Sci. 2020275, 102061. DOI: 10.1016/j.cis.2019.102061

Amaral, P.F.; Coelho, M.A.Z.; Marrucho, I.M.; Coutinho, J.A. Biosurfactants, 1st ed.; Sen, R., Ed.; Springer, New York, NY, 2010; 672, pp. 236-249.

(3) The authors tried many techniques such as FT-IR, NMR, GC-MS, X ray diffraction, biochemical and many to determine the structure. Unfortunately, the authors did not propose the structure of SE which is expected by the readers.

Thank you for your comment. Herein, we tried to gather as much structural and physicochemical information as possible of our obtained product, and we were able to conclude that it corresponds to a complex lipid/protein/carbohydrate macromolecular structure. In general, considering the vast majority of published studies within the field, the characterization of such biocompounds is limited to simple spectroscopic and chromatographic techniques, as it is the case of the work of Gil et al. (2022), Kourmentza et al. (2019) and Silva et al. (2022), that reported similar analyses for analogous molecules. The authors could only point out some chemical functions and their correlations, without determining the precise molecular structure of the compounds. In such cases, a thorough and more detailed investigation is needed to present a precise final molecular structure of the active ingredient that is responsible for the emulsifying activity observed, which fell outside the scope of our present contribution.

Kourmentza, C., Araujo, D., Sevrin, C., Roma-Rodriques, C., Ferreira, J. L., Freitas, F., ... & Reis, M. A. (2019). Occurrence of non-toxic bioemulsifiers during polyhydroxyalkanoate production by Pseudomonas strains valorizing crude glycerol by-product. Bioresource technology281, 31-40.

Gil, C. V., Rebocho, A. T., Esmail, A., Sevrin, C., Grandfils, C., Torres, C. A., ... & Freitas, F. (2022). Characterization of the Thermostable Biosurfactant Produced by Burkholderia thailandensis DSM 13276. Polymers14(10), 2088.

Silva, T. P., Paixão, S. M., Tavares, J., Gil, C. V., Torres, C. A., Freitas, F., & Alves, L. (2022). A New Biosurfactant/Bioemulsifier from Gordonia alkanivorans Strain 1B: Production and Characterization. Processes10(5), 845.

(4) Check spacings; No abbreviation (BE) in the title or subtitle; All the figure numbers follow the numerological order. Figure 1B --> Figure 1A; Move the y axis title to the left.

Thank you for the careful reading of the manuscript. In the new version, those and other minor errors were corrected.

Reviewer 2 Report

I acknowledge the importance to  seek and find new biosurfactants for the future applications in cosmetics/ chemical industry/ agriculture/ food and feed industry. 

The authors of the article have isolated molecules that posess some biosurfactant activity extracellular fraction of Scheffersomyces shehatae 16-BR6-2AI  fermentation.

However, eventhough the fraction from fermentation has some biosurfactant activity, it’s molecular structure is not clear. Also it is not clear if the fraction contain single (biosurfactant) molecule or it is just mixture of unrelated molecules which all together act as surfactant. Therefore, I have several questions, comments and suggestions. 

Can You modelate / explain in details – is this a single molecule with specific structure of (glycolipid/ peptidolipid anything else)?

Have You tried TLC (thin layer chromatography), do the extract contain single molecule or is it mixture of lipids / proteins/ carbohydrates with similar hydrofobicity (therefore they have been extracted together as a single fraction)?

Have You treated Your extracts with hexane or any other highly non-polar solvent to wash the lipid/s away?

Have You tried to digest peptides of the extracted biosurfactants by some proteases and do the activity of the surfactant still remain?

Author Response

(1) I acknowledge the importance to seek and find new biosurfactants for the future applications in cosmetics/ chemical industry/ agriculture/ food and feed industry. The authors of the article have isolated molecules that posess some biosurfactant activity extracellular fraction of Scheffersomyces shehatae 16-BR6-2AI fermentation. However, eventhough the fraction from fermentation has some biosurfactant activity, it’s molecular structure is not clear. Also it is not clear if the fraction contain single (biosurfactant) molecule or it is just mixture of unrelated molecules which all together act as surfactant. Therefore, I have several questions, comments and suggestions. Can You modelate/explain in details – is this a single molecule with specific structure of (glycolipid/ peptidolipid anything else)?

Thank you for acknowledging the relevance of the study. The biochemical and spectroscopic analyzes of the obtained product that were carried out herewith led to important structural and physicochemical information, that was enough to state that it corresponds to a complex lipid/protein/carbohydrate macromolecular structure. However, a deeper investigation to assess the specific and precise chemical structure, as well as the intra- and intermolecular interactions that are present stabilizing the supramolecular configuration, falls outside the scope of our present initiative. Therefore, considering the set of results we obtained, we cannot say whether it represents a single compound, or a mixture of carbohydrates, proteins, and lipids associated by strong non-covalent forces. The analyzes chosen to be performed herewith were based on analogous studies related to polymeric bioemulsifiers, such as those of Gil et al. (2022), Kourmentza et al. (2019), and Silva et al. (2022), who carried out structural and physicochemical characterization by means of FTIR, NMR, GC-MS, X-ray diffraction and biochemical analyses. In the three cases, the molecules were characterized as polymers without specifying, however, if consisted of a single molecule or on a mixture.

Kourmentza, C., Araujo, D., Sevrin, C., Roma-Rodriques, C., Ferreira, J. L., Freitas, F., ... & Reis, M. A. (2019). Occurrence of non-toxic bioemulsifiers during polyhydroxyalkanoate production by Pseudomonas strains valorizing crude glycerol by-product. Bioresource technology281, 31-40.

Gil, C. V., Rebocho, A. T., Esmail, A., Sevrin, C., Grandfils, C., Torres, C. A., ... & Freitas, F. (2022). Characterization of the Thermostable Biosurfactant Produced by Burkholderia thailandensis DSM 13276. Polymers14(10), 2088.

Silva, T. P., Paixão, S. M., Tavares, J., Gil, C. V., Torres, C. A., Freitas, F., & Alves, L. (2022). A New Biosurfactant/Bioemulsifier from Gordonia alkanivorans Strain 1B: Production and Characterization. Processes10(5), 845.

(2) Have You tried TLC (thin layer chromatography), do the extract contain single molecule or is it mixture of lipids / proteins/ carbohydrates with similar hydrofobicity (therefore they have been extracted together as a single fraction)?

Thank you for your question. Due to the high molecular weight of the product, TLC was not a useful technique to promote separation and, therefore, to indicate the presence of more than one chemical structure.

(3) Have You treated Your extracts with hexane or any other highly non-polar solvent to wash the lipid/s away?

As one of the nutritional sources used consisted of soybean oil, after the fermentation process, the samples were treated with hexane before centrifugation. The complete procedure is described in the experimental part (section 2.5).

(4) Have You tried to digest peptides of the extracted biosurfactants by some proteases and do the activity of the surfactant still remain?

Thank you for the question. The process was actually carried out, with the bioemulsifier being treated with proteinase K, followed by emulsification test. The process led to reduced emulsifying action, with a loss of emulsion stability 24 hours. Bhaumik et al. (2020) suggested that the protein fraction present in such molecules plays an important role in the corresponding action of emulsification.

Reviewer 3 Report

The manuscript describes the production and physical, chemical, and structural characterization of the bioemulsifier secreted by the yeast Scheffersomyces shehatae 16-BR6-2AI in a medium containing hemicellulosic sugarcane bagasse hydrolysate combined with soybean oil. Studies of this nature are welcome in the context of the expanding bioeconomy. In this economic paradigm, biorefineries play a key role.

The paper is well-written and very well-organized. The introductory section adequately contextualizes the relevance of the topic and presents the objective of the study as being "to produce and characterize the BE produced by the yeast Scheffersomyces shehatae 16-BR6-2AI in a medium containing sugarcane bagasse hemicellulose hydrolysate combined with soybean oil in the context of biorefineries." The materials and methods section is well structured, detailing the materials used and the laboratory procedures used for the preparation, production and isolation of the bioemulsifier (BE), the biochemical, physicochemical and structural characterization of the BE. In section 3, the main results of the analyses described in the method are presented and discussions are made in comparison to other results reported in the literature for alternative BEs. The authors conclude that BE presents potential applications as an emulsifying agent in different industrial sectors. Finally, the list of references used is extensive, relatively up-to-date, and appropriate to the topic developed in the study. In general, the manuscript presents great consistency and interconnection between the parts.

I have not identified any critical points that require mandatory adjustments in my evaluation.

Three minor points that authors may possibly consider:

a. Many of the keywords are present in the title and abstract and could be replaced by other terms that would broaden the scope of search and retrieval of the article in search engines.

b. In line 85, the authors stated that BE is a sustainable alternative obtained by integrating two biorefineries. It would be timely if this integration and these two biorefineries were clearer.

c. The authors could consider including a list of abbreviations and acronyms.

Author Response

(1) The manuscript describes the production and physical, chemical, and structural characterization of the bioemulsifier secreted by the yeast Scheffersomyces shehatae 16-BR6-2AI in a medium containing hemicellulosic sugarcane bagasse hydrolysate combined with soybean oil. Studies of this nature are welcome in the context of the expanding bioeconomy. In this economic paradigm, biorefineries play a key role. The paper is well-written and very well-organized. The introductory section adequately contextualizes the relevance of the topic and presents the objective of the study as being "to produce and characterize the BE produced by the yeast Scheffersomyces shehatae 16-BR6-2AI in a medium containing sugarcane bagasse hemicellulose hydrolysate combined with soybean oil in the context of biorefineries." The materials and methods section is well structured, detailing the materials used and the laboratory procedures used for the preparation, production and isolation of the bioemulsifier (BE), the biochemical, physicochemical and structural characterization of the BE. In section 3, the main results of the analyses described in the method are presented and discussions are made in comparison to other results reported in the literature for alternative BEs. The authors conclude that BE presents potential applications as an emulsifying agent in different industrial sectors. Finally, the list of references used is extensive, relatively up-to-date, and appropriate to the topic developed in the study. In general, the manuscript presents great consistency and interconnection between the parts.

I have not identified any critical points that require mandatory adjustments in my evaluation.

Thank you for the positive comments about our manuscript. All suggestions were considered in the new version of the manuscript.   

(2) Many of the keywords are present in the title and abstract and could be replaced by other terms that would broaden the scope of search and retrieval of the article in search engines.

Thank you very much, the suggestion was addressed in the new version of the manuscript.

(3) In line 85, the authors stated that BE is a sustainable alternative obtained by integrating two biorefineries. It would be timely if this integration and these two biorefineries were clearer.

Thank you very much, the suggestion was addressed in the new version of the manuscript.

(4) The authors could consider including a list of abbreviations and acronyms.

Thank you for the suggestion. The list of abbreviations presented below was included in the new version of the manuscript.

PCBs: polychlorinated biphenyls

PAHs: polynuclear aromatic hydrocarbons

GRAS: Generally Recognized As Safe

BE: Bioemulsifiers

C-5: 5 carbon carbs

C-6: 6 carbon carbs

SBHH: Sugarcane bagasse hemicellulosic hydrolysate

HPLC: High-Performance Liquid Chromatography

YMA: Yeast Malt Agar

EI24: Emulsifying index

ST: Surface tension

PEO: Polyethylene oxide 

HMF: Hydroxymethylfufural

Mw: Average molecular mass

Tm: Melting temperature

CMC: Critical micelle concentration

FTIR: Fourier transform infrared spectroscopy

CG-MS: Gas chromatograph-mass spectrometer

TGA: Thermos gravimetric analysis

DSC: Differential Scanning Calorimetric

1H NMR: 1H Nuclear magnetic resonance

13C NMR: 13C Nuclear magnetic resonance

YX/S: yield coefficient of cell growth per substrate

YP/S: Product yield coefficient per substrate

BSA: Bovine Serum Albumin

GPC: Gel Permeation Chromatography 

Round 2

Reviewer 2 Report

Dear authors, thank You for Your answers and comments to my questions. I have no more objections/ other suggestions to add.   

a note: please, change "Biemulsifier" to "bioemulsifier" in all manuscript.